# Detection of Avian Influenza Virus in Pigeons

**DOI:** 10.3390/v17040585

**Published:** 2025-04-18

**Authors:** Ning Cui, Peipei Wang, Qinghua Huang, Zihao Yuan, Shuai Su, Chuantian Xu, Lihong Qi

**Affiliations:** 1Institute of Animal Science and Veterinary Medicine, Shandong Academy of Agricultural Sciences, Jinan 250100, China; 2Key Laboratory of Livestock and Poultry Multi-Omics of MARA, Jinan 250100, China; 3Tiankang Bio-Pharmaceutical Co., Ltd., Wulumuqi 830011, China; 4Shandong Provincial Key Laboratory of Animal Biotechnology and Disease Control and Prevention, College of Veterinary Medicine, Shandong Agricultural University, Tai’an 271000, China

**Keywords:** avian influenza virus, pigeon, subtype, evolution

## Abstract

Pigeons (Columba livia) are usually kept as free-ranging or racing birds, and they have direct contact with livestock, poultry, and humans. Therefore, they may have an important role in the ecology of influenza virus among various species. In the present study, we bring together all available sequence data of pigeon avian influenza virus (AIV) from public databases to address the current understanding of the genomic characteristics and emergence of each subtype of AIV in pigeons. Collectively, we identified 658 pigeon AIV strains in 21 countries across the world, which were mainly distributed in Europe, Asia, and North America. H1 (2), H2 (1), H3 (8), H5 (71), H6 (16), H7 (16), H9 (543), and H11 (1) AIV subtypes have been identified in pigeons. In addition, we interrogate features of the H5, H6, H7, and H9 subtypes of pigeon AIV, which are relatively common in pigeons. It is particularly noteworthy that the H5 AIV strains identified in pigeons are all classified as HPAIV. For the first time, this study presents a complete overview of the multiple AIV subtypes that have been circulating in pigeons, providing information on their distribution and genomic characteristics. This study will help to understand the molecular evolution of AIV in pigeons.

## 1. Introduction

Avian influenza virus (AIV) belongs to the family of Orthomyxoviridae and is a multi-host virus affecting wildlife, domestic animal, and human health [1]. The viral genome of AIV contains a single-stranded ribonucleic acid (RNA) genome composed of eight gene segments encoding at least 11 viral proteins [2]. Currently, AIV is categorized based on the antigenic properties of the surface glycoproteins, HA and NA, into 19 and 11 subtypes, respectively [3]. Furthermore, AIV can be categorized as low pathogenic AIV (LPAIV) or highly pathogenic AIV (HPAIV) based on the pathogenicity in chickens upon experimental exposure and scored on an intravenous pathogenicity index [4]. Currently, the most common method to identify its pathogenicity is by sequencing the HA gene segment to characterize the presence of multiple basic amino acids at the HA cleavage site (HACS) [4]. The HACS motif of HPAIV typically contains five or more basic consecutive amino acids, facilitating cleavage by the ubiquitously expressed proprotein convertases, which is the prime determinant of the high infectivity and pathogenicity of HPAIV. To date, HPAIV has been restricted to H5 and H7 subtypes of AIV only [3].

Pigeons (Columba livia) are a common poultry species in many countries and likely to be regularly exposed to AIV; they can be considered a potential ecological niche [5]. Field surveys of pigeons often show low frequencies and relatively low seroprevalence, and experimental studies have generally demonstrated high resistance and limited susceptibility to multiple strains of AIV, especially for LPIAV [6,7]. Nonetheless, AIV is a naturally occurring virus in pigeons, and it is consistently detected in surveillance schemes. In fact, most subtypes of AIV have been identified in pigeons, and some individuals and some strains can lead to productive infections and transmission [8,9,10]. Notably, pigeons are usually raised as free-ranging or racing birds, and they frequently interact with poultry, livestock, and even humans. Therefore, they may also play a significant role in the interspecies ecology of influenza viruses. In this study, we summarize and address the current understanding of the evolution and emergence of each subtype of AIV in pigeons. In particular, we interrogate features of the H5, H6, H7, and H9 subtypes of pigeon AIV strains, which are relatively common in pigeons.

## 2. Materials and Methods

### 2.1. Sequence Collection

All previously published sequences of pigeon AIV strains up to date as of 30 December 2024 were downloaded from the Global Initiative on Sharing Avian Influenza Data (GISAID, www.gisaid.org) and the Influenza Virus Resource at the National Center for Biotechnology Information (accessed on 30 December 2024). All replicate submissions were removed by identifying sets of isolates with identical sequences for all segments, with each isolate containing one to eight segments. Sequences’ data lists and detailed information regarding the previously published pigeon AIV strains are summarized in the Appendix A.

### 2.2. Phylogenetic and Molecular Analyses

DNA sequences were compiled and edited using the Lasergene sequence analysis software package (DNASTAR, version 7 Madison, WI, USA), and multiple sequence alignment was carried out using ClustalW (version 2.0.10). The phylogenetic analysis of each viral gene segment was performed through the neighbor-joining method with 1000 bootstrap replicates using MEGA (version 5.05). The amino acid sequences of encoding proteins were derived from DNA sequences and subsequently aligned using ClustalW to identify conservation and variations in the amino acid sequences.

### 2.3. Genotypic Analysis

The genotypes of H5 and H9 subtypes of pigeon AIV strains were determined through the combination of clade assignments of each of the eight segments in accordance with the genotype naming scheme used in previous studies [11,12]. A genotype was defined when the phylogenetic lineages of the eight gene segments resulted in a unique gene grouping or constellation.

## 3. Results and Discussion

### 3.1. Distributions and Subtypes of Pigeon AIV Strains

For the global analysis, we downloaded all of the available gene sequences of pigeon AIV strains from the public databases. Pigeon AIV was first reported in 1981 and has been reported every year since 2000. The number of reported strains of pigeon AIV has risen sharply since 2012 (Figure 1A). To date, 658 pigeon AIV strains have been reported in 21 countries worldwide, which are mainly distributed in Europe, Asia, and North America (Appendix A) (Figure 1B). Collectively, the H1 (2), H2 (1), H3 (8), H5 (71), H6 (16), H7 (16), H9 (543), and H11 (1) subtypes of AIV strains have been identified in pigeons. The H9 and H5 subtypes of AIV strains are the most abundant subtypes reported in pigeons (Figure 1C).

### 3.2. H1 Virus

Avian-origin H1 subtypes of influenza viruses caused the human pandemics in 1918 [13] and 2009 [14], and the later strain that caused the outbreak was derived from a triple recombinant swine influenza virus circulating in pigs and a Eurasian swine influenza virus. The H1 subtype of AIV in combination with N1 and N2 has been identified in pigeons (Appendix A). A/pigeon/MN/1407/1981(H1N1) was reported in USA and A/pigeon/Zhejiang/1120087/2014(H1N2) was reported in China. The phylogenetic analysis showed that all genes of A/pigeon/MN/1407/1981(H1N1) were clustered in the North American lineage and all genes of A/pigeon/Zhejiang/1120087/2014(H1N2) were clustered in the Eurasian lineage [15].

### 3.3. H2 Virus

The circulation of the H2 subtype of AIV in domestic animals also increases the risk of human exposure to these viruses. For example, the H2N2 influenza virus that caused the 1957 Asian pandemic derived its genes from human H1N1 and avian H2N2 viruses [16]. A/Pigeon/Longquan/LQ67/2016(H2N8) has been identified in pigeons in China and showed the highest homology with the reassorted H2N8 virus from domestic ducks A/duck/Zhejiang/6D10/2013(H2N8), both of which were clustered in the Eurasian lineage [17] (Appendix A).

### 3.4. H3 Virus

The H3 subtype of AIV can provide genes for human influenza virus through gene reassortment, which has raised great concerns about its potential threat to human health. For example, the A/H2N2 strain circulated seasonally among humans for approximately 11 years after the “Asian flu” from 1957, and it recombined with the H3 avian virus, resulting in the emergence of the novel H3N2 virus until 1968 [18]. Collectively, eight H3 subtypes of AIV strains have been identified in pigeons since 2000, including one N2 subtype (A/pigeon/Guangxi/128P9/2012), one N3 subtype (A/Pigeon/Nanchang/9-058/2000), three N6 subtypes (A/Pigeon/Nanchang/8-142/2000, A/Pigeon/Nanchang/11-045/2000, and A/pigeon/Guangxi/020P/2009), two N8 subtypes (A/pigeon/Anhui/08/2013 and A/Rufous Turtle Dove/South Korea_KNU18-19/2018), and one undefined subtype (A/pigeon/Germany/R42/01) (Appendix A). The analysis of the nucleotide sequence of A/pigeon/Guangxi/020P/2009(H3N6) indicated that both HA and NA genes belong to the Eurasian lineage. Its other internal genes are closely related to H3N8, H4N6, H6N2, H3N2, and H4N2 subtypes of AIVs, which suggests that this H3N6 strain went through extensive reassortment with different subtypes of AIV isolates [19]. Similarly, our analysis of the sequence of the remaining H3 subtypes of pigeon AIV strains indicated that they might all be reassortment viruses with segments from different sources.

### 3.5. H5 Virus

HPAI H5N1 is the most frequent subtype of AIV that causes death in many avian species and mammals, including humans [3,20]. A total of 71 H5 subtypes of AIV strains have been identified in pigeons since 2001, including 54 H5N1 viruses, 4 H5N6 viruses, 12 H5N8 viruses, and 1 undefined H5 virus (Appendix A). According to the amino acid sequence deduced from the HA gene segments, all of the pigeon H5 subtypes of AIV strains harbor more than five basic amino acids in the pHACS motifs of the HA precursor protein, suggesting that they all belong to HPAI (Table 1). We inferred phylogenetic trees for the HA and NA gene segments. The HA phylogenetic tree was inferred using 65 HA sequences, including 49 HA sequences from H5N1 viruses, 4 HA sequences from H5N6 viruses, and 12 HA sequences from H5N8 viruses (Figure 2A). The NA phylogenetic tree was inferred using 37 NA sequences, and it clearly displayed the three NA subtypes that occur in combination with the H5 gene in pigeons (Figure 2B). Phylogenetic analysis showed that most of the HA genes of the H5N1 strains from pigeons (Figure 2A) were distributed in the branch of Clade 1 (10 isolates) and Clade 2 (37 isolates), except for the first H5 strain A/Pigeon/Hong Kong/SF215/01 and the A/pigeon/Huadong/QPG/2010 strain, which were located in Clade 4 and Clade 7.2, respectively. The H5N1 virus was first identified pigeons in 2001. All of the 10 early viruses isolated before 2005 were clustered in Clade 1, and all of the 6 later isolates after 2020 were clustered in Clade 2.3.4.4b. The remaining isolates between 2005 and 2020 belong to Clade 2.1 (3 isolates), Clade 2.2 (17 isolates), Clade 2.3.1 (1 isolates), Clade 2.3.2.1a (1 isolates), Clade 2.3.2.1b (1 isolates), Clade 2.3.2.1c (3 isolates), and the 2.3.4 Clade (5 isolates). A total of 22 nearly complete H5N1 viral genomes were available. We then inferred phylogenetic trees for each of the six internal gene segments, and, on the basis of clade classification of the internal genes, they were divided into four early genotypes (EG) of H5N1 viruses and five later reported H5N1 genotypes (G) in accordance with the genotype naming scheme used in a previous study [11] (Figure 2B). The viruses in genotypes 18 and 19 are associated with human infections [21,22]. Phylogenetic analysis showed that the HA genes of four H5N6 strains from pigeons were clustered in Clades 2.3.2.1c, 2.3.4.4h, and 2.3.4.4e, respectively, while the HA genes of 12 H5N8 strains from pigeons were all clustered in Clade 2.3.4.4b (Figure 2A). On the basis of clade classification of the internal genes, we observed that H5N6 strains were divided into two genotypes and H5N8 strains were divided into three genotypes (Figure 2B). The results suggested that Clade 2.3.4.4b included the dominant H5 HPAIV strains were prevalent in pigeons. Indeed, H5 Clade 2.3.4.4.b led to a massive number of outbreaks worldwide in wild and domestic birds, and it has evolved, adapted, and spread to species other than birds, with potential mammal to mammal transmission [23,24]. For example, genomic analysis revealed 99.7% nucleotide identity between H5N1 viruses circulating in pigeon flocks and those infecting dairy cattle during the 2023 US outbreak [25]. Considering the high risk of morbidity and widespread transmission of the H5 subtype of AIV, especially the H5 Clade 2.3.4.4.b, epidemiological investigation and immunization implementation for the virus in pigeons should also be emphasized.

### 3.6. H6 Virus

The H6 subtype of AIV was first isolated from a turkey in 1965 and subsequently from shorebirds and wild ducks [26,27]. Currently, the H6 AIV has become one of the most abundantly detected subtypes circulating in wild birds and domestic poultry throughout different continents in the world [28,29]. The H6 subtype of AIV strains has a broader host range than any other subtype and could sporadically infect humans [30]. We inferred phylogenetic trees for HA gene fragments using 238 full-length or nearly full-length H6 genes of representative viruses retrieved from public databases according to previously defined H6 groups [31] (Figure 3). According to the phylogenetic analysis of the H6 gene fragments, they were classified into the gene pools of the American lineage and the Eurasian lineage. The Eurasian lineage was further divided into three major branches (Clades 1–3). Collectively, 17 H6 subtypes of AIV strains have been identified in pigeons since 2003, including one N1 subtype, six N2 subtypes, and nine N6 subtypes (Appendix A). Phylogenetic analyses of HA genes demonstrated that all pigeon AIV strains were classified into the Eurasian lineage. Two H6N2 subtypes of pigeon AIV strains were classified into Clade 1, which contains mainly waterfowl-derived AIV isolates. A/pigeon/Hong Kong/WF47/2003 (H6N1) was classified into Clade 2, which contains AIV strains isolated from wild birds. Two H6N2 subtypes of pigeon AIV and all H6N6 subtypes of pigeon AIV strains were classified into Clade 3, which contains AIV strains derived from poultry (mainly ducks) and environments. This suggests that infection of the H6 subtype of pigeon AIV is closely related to wild birds and waterfowl.

### 3.7. H7 Virus

H7 influenza viruses are able to infect a broad range of species, from an array of wild bird species to poultry and mammals, including seals, pigs, horses, and humans [32]. In poultry, LPAIV H7 can circulate asymptomatically and evolve into HPAIV H7, causing severe systemic diseases and mortality [33]. In addition, humans have been shown to become infected sporadically with either LPAIV or HPAIV H7 viruses. It is generally considered that H7 viruses have lower virus excretion and a limited ability to spread in pigeons [34]. Large-scale epidemiological studies have identified the presence of H7N9 viruses in asymptomatic pigeons, bringing these viruses back into the spotlight due to their potential role as an interspecies bridge in avian influenza ecology [35]. Collectively, 16 H7 subtypes of AIV strains have been identified in pigeons since 2003, including 1 N2 subtype, 3 N7 subtypes, 1 N8 subtype, and 11 N9 subtypes (Appendix A). All viruses were isolated from Yangtze River Delta regions and nearby southern areas, including six strains in Zhejiang Province, four strains in Jiangsu Province, three strains in Shanghai Province, two strains in Jiangxi Province, and one strain in Fujian Province (Figure 4A). According to the amino acid sequence deduced from the HA nucleotide sequences, all of the pigeon H7 subtypes of AIV strains harbor two basic amino acids in the pHACS motifs, suggesting that they all belong to LPAIV, which might account for the lower virus excretion and limited spreading ability in pigeons. According to the previous lineage naming scheme [36], we observed that the H7N2 subtype of the pigeon AIV strain was classified into H7N9 lineage Wave1 (Figure 4B). Three H7N7 subtypes of pigeon AIV strains all belong to the H7N7 lineage, with one duck-derived strain and two chicken-derived strains. The H7N8 subtype of the pigeon AIV strain was classified into the H7N9 lineage Wave2-C. Of the H7N9 subtypes of pigeon AIV strains, nine strains belong to the H7N9 lineage Wave1 and two strains belong to W2-C. In September 2017, vaccination of poultry was initiated using a bivalent H5/H7 vaccine, and the H7 subtype of AIV has not been detected in pigeons since then, demonstrating that the existing H7N9 vaccine candidate could protect against divergent H7 subtypes of AIV strains.

### 3.8. H9 Virus

H9 influenza virus was first isolated from turkeys in North America in 1966 [37], and it has become the most prevalent subtype of AIV in multiple avian species [35,38,39]. In pigeons, H9N2 AIV is also the most prevalent subtype, as 543 strains have been reported, including 77 whole viral genomes (Appendix A) and 465 partial viral gene fragments. According to the defined criterion based on the phylogenetic analysis of eight AIV fragments used in a previous study [12] (Figure 5A), we observed that 14 genotypes were defined for the 77 H9N2 viral whole genomes, including 8 reported stable or transient genotypes and 6 novel genotypes. Forty-two strains, including A/Rock Pigeon/Vietnam/6/2009, A/pigeon/South_Korea/N23-036/202340, and Chinese H9N2 AIV strains from different provinces, were classified under the H9N2 AIV G57 genotype, which is the dominant genotype in pigeons, as reported in other poultry species. G49, prevalent in Chinese Ningxia, and G68, prevalent in Chinese Guizhou, were also the major stable H9N2 AIV genotypes in pigeons, which contain eight and six strains, respectively. Two strains from the Chinese Jiangsu Province were classified into the transient G115 genotype. Four strains from Chinese Hong Kong were classified into the G4, G51, G52, and G63 H9N2 AIV genotypes, respectively. The remaining strains were classified into six novel genotypes, among which three genotypes contain strains from Guangdong Province, Fujian Province, and Anhui Province. Two strains from South Asia, including A/pigeon/Pakistan/25A/2015 and A/pigeon/Bangladesh/4303/2009, were classified into one novel genotype. The pigeon H9N2 AIV strains from Egypt were classified into two novel genotypes, with the later genotype derived from the earlier genotype, and it gained a novel NP fragment. The results suggested that the distribution of the H9N2 subtype of pigeon AIV possessed regional and spatiotemporal characteristics. In addition, a pigeon H9N6 subtype AIV strain A/pigeon/Fujian/9.25_FZHX0007-C/2017 has been reported, which might be re-contaminated by gaining an NA fragment from the duck-derived H6N6 subtype of AIV.

### 3.9. H11 Virus

The H11 influenza virus was first identified in 1956. In stark contrast to the H5, H6, H7, and H9 subtypes of AIV strains, the H11 subtype of the virus has limited host range, as waterfowl and shorebirds are the main reservoirs, while few H11 virus isolates have been identified in poultry [40]. An H11N2 subtype of the pigeon AIV strain A/pigeon/Shandong/D173/2019 has been identified in pigeons from a live poultry market (Appendix A), and a phylogenetic tree showed that its HA gene was clustered into the Eurasian lineage, formed mainly by the viruses detected in migratory birds and ducks. Phylogenetic analysis of the internal genes revealed that the pigeon H11N2 strain is a reassortant virus containing the M gene derived from H5N6 viruses that have been detected in birds and mammals. It is worth noting that the pigeon H11N2 strain has adapted to replicate and transmit in chickens, demonstrating that it poses a great risk of infection in poultry, especially in chickens [40].

### 3.10. Mixed Viruses

The mixed infection of AIV provides the necessary prerequisite and opportunity for viral gene recombination to generate novel genotype viruses. Mixed infection of different subtypes of AIV strains has also been identified in pigeons (Appendix A). A/pigeon/Fujian/1.17_FZHX0111-C/2017 contains mixed viruses of the H6N6 and H7N8 subtypes. A/pigeon/Jiangxi/10.19_NCDZT62A4-OC/2018 contains mixed viruses of H6N2 and others. A/pigeon/Jiangxi/JXA130010/2013 is a mixed virus of the H7, H9, N2, and N9 subtypes. Combined with the aforementioned statistical data on pigeon AIV, pigeons can serve as “virus blenders” capable of simultaneous infection with multiple AIV subtypes. Such mixed infections significantly promote genetic recombination, leading to the emergence of novel strains and increasing pandemic risks.

## 4. Conclusions

Overall, we demonstrate that the H1, H2, H3, H5, H6, H7, H9, and H11 subtypes of AIV strains have been identified in pigeons. H9 and H5 AIV are the most abundant subtypes reported in pigeons. It is particularly noteworthy that the H5 AIV strains identified in pigeons are all classified as HPAIV. Therefore, it is imperative to pay close attention to epidemiological investigation and immunization implementation for pigeons. For the first time, this study presents a complete overview of the multiple AIV subtypes that have been circulating in pigeons, providing information on their distribution and genomic evolution. As most AIV subtypes do not cause obvious symptoms in pigeons and only limited surveillance has been conducted, the prevalence of AIV, especially LPAIV in pigeons, is likely to be higher than we document here. From the viewpoint of the sequence analysis of pigeon AIV, free-ranging pigeons always play a negligible but non-zero role as an interspecies bridge in the ecology of influenza virus dynamics.

## Figures and Tables

**Figure 1 viruses-17-00585-f001:**
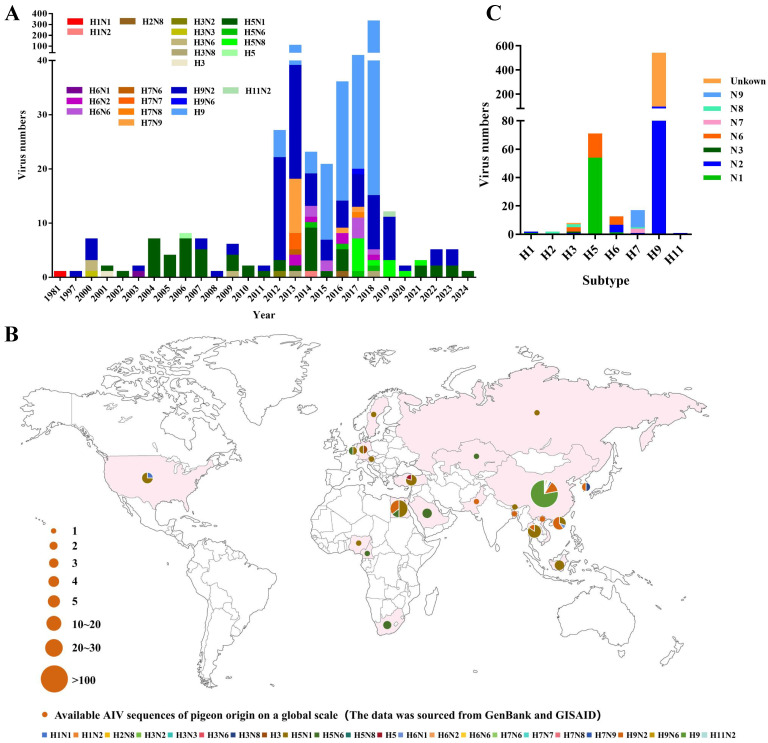
Global prevalence of pigeon AIV. (**A**) Subtypes and temporal distribution of pigeon AIV. The temporal distribution of the isolates of each subtype is indicated with different colors of tips. (**B**) Global geographical distribution of different subtypes of pigeon AIV strains. The proportion of isolates for each subtype category is visualized in a pie chart. The size of the pie chart is proportional to the number of isolates in each continent. (**C**) Summary of different subtypes of pigeon AIV strains. The y axis denotes the number of strains. All of the public data used in this study were up to date as of 30 December 2024.

**Figure 2 viruses-17-00585-f002:**
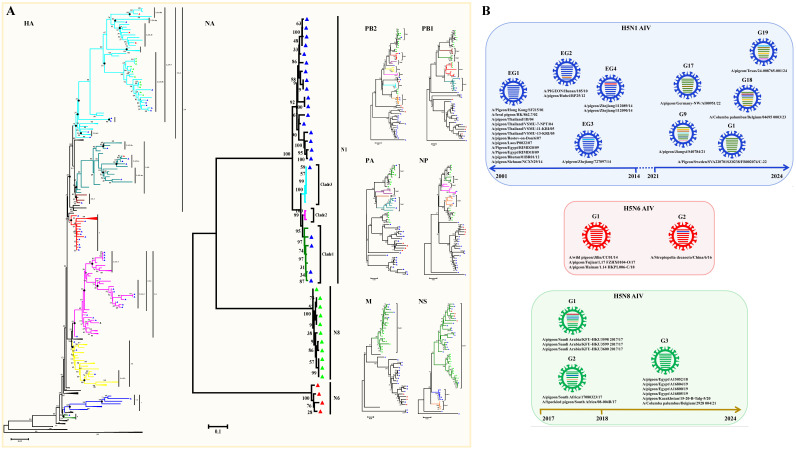
Genomic Characterization of the H5 subtype of pigeon AIV. (**A**) Phylogenetic analyses of the H5 subtype of pigeon AIV. The phylogenetic trees for HA, NA, PB2, PB1, PA, NP, M, and NS segments were generated through the neighbor-joining method with MEGA (version 5.05). The numbers at the branch points show the bootstrap values calculated from 1000 bootstrap replicates, and the scale bars indicate the number of nucleotide substitutions per site. The color of the branches indicates different clades. The clades and the assigned numbers are labeled separately. The H5N1, H5N6, and H5N8 subtypes of pigeon AIV strains are labeled by blue, red, and green triangles. (**B**) Genotype analysis of the H5 subtype of pigeon AIV. Viral genotypes were analyzed for the H5 subtype of pigeon AIV based on gene phylogenic analysis and determined through the combination of clade assignments of each of the eight segments (**A**) according to the previous study. The eight gene segments (horizontal bars from the top) are arranged by PB2, PB1, PA, HA, NP, NA, M, and NS in order of size. Each color corresponds to a viral segment clade.

**Figure 3 viruses-17-00585-f003:**
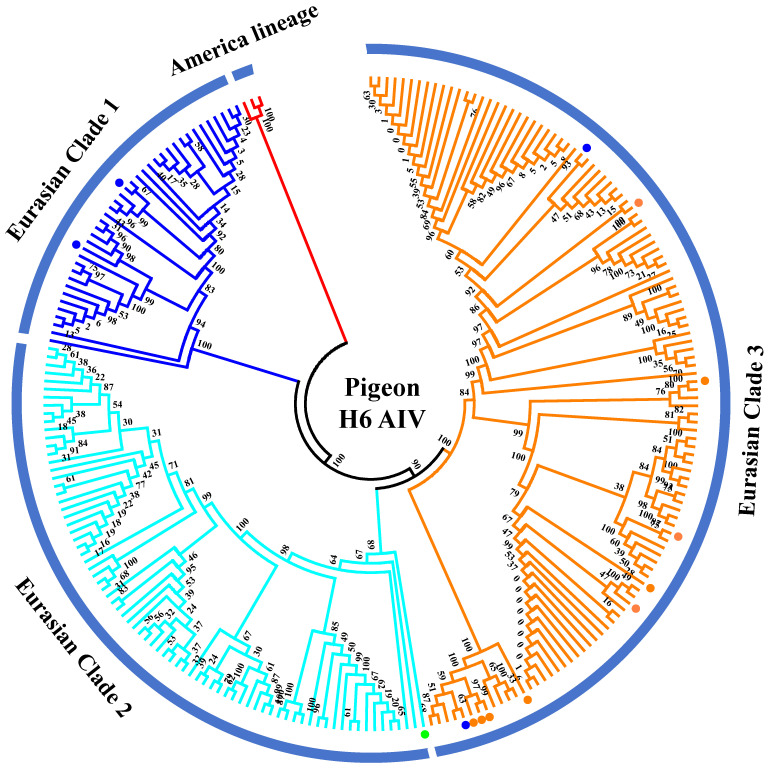
Phylogenetic analyses of the H6 subtype of pigeon AIV. Phylogenetic trees for HA gene fragments were generated using 238 full-length or nearly full-length H6 genes retrieved from public databases. The H6N1, H6N2, and H6N6 subtypes of pigeon AIV strains are labeled by green, blue, and orange dots.

**Figure 4 viruses-17-00585-f004:**
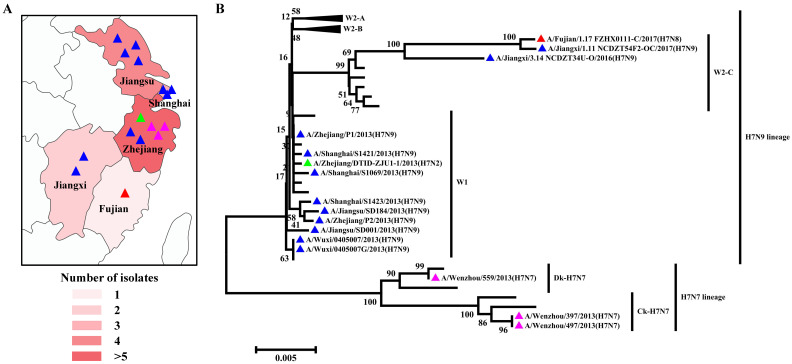
Distribution and Phylogenetic analyses of the H7 subtype of pigeon AIV. (**A**) Geographical distribution of the H7 subtype of pigeon AIV in the Yangtze River Delta and nearby areas in China. The colors’ depth represents the number of strains. (**B**) The phylogenetic tree of the HA genes of the H7 subtype of pigeon AIV. The H7N2, H7N7, H7N8, and H7N9 subtypes of pigeon AIV strains are labeled by green, pink, red, and blue triangles.

**Figure 5 viruses-17-00585-f005:**
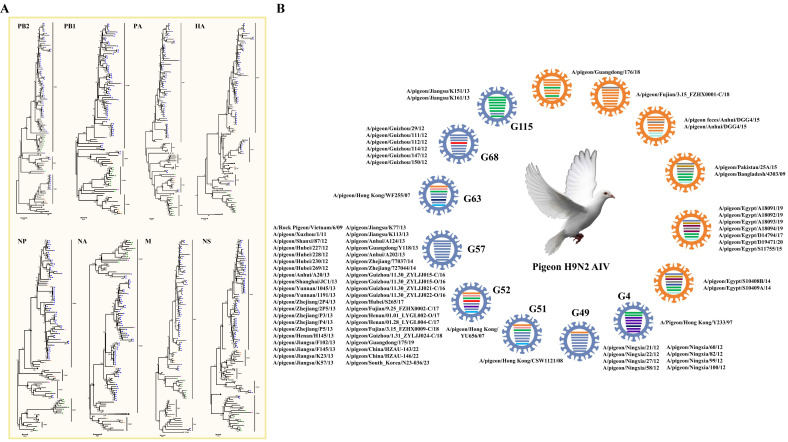
Genomic Characterization of the H9 subtype of pigeon AIV. (**A**) Phylogenetic analyses of the H9 subtype of pigeon AIV. The phylogenetic trees for HA, NA, PB2, PB1, PA, NP, M, and NS segments were generated through the neighbor-joining method with MEGA (version 5.05). The numbers at the branch points show the bootstrap values calculated from 1000 bootstrap replicates, and the scale bars indicate the number of nucleotide substitutions per site. The clades and the assigned numbers were labeled separately. Colored triangles represent pigeon H9 AIV strains from different areas. (**B**) Genotype analysis of the H9 subtype of pigeon AIV. Viral genotypes were analyzed for the H9 subtype of pigeon AIV strains based on gene phylogenic analysis and determined through the combination of clade assignments of each of the eight segments (**A**) according to the previous study. The eight gene segments (horizontal bars from the top) are arranged by PB2, PB1, PA, HA, NP, NA, M, and NS in order of size. Each color corresponds to a viral segment clade.

**Table 1 viruses-17-00585-t001:** Haemagglutinin cleavage site motif of the H5 subtype of pigeon AIV strains.

Isolate/Name	Subtype	Consensus HACS Motif	Number of Basic Amino Acids in HACS Motif
A/Pigeon/Hong Kong/SF215/01	H5N1	PQ_RERRRKKR/G	6
A/feral pigeon/HK/862.7/2002	H5N1	PQ_IERRRKKR/G	6
A/pigeon/Thailand/KU-03/04	H5N1	PQ_RERRRKKR/G	6
A/pigeon/Thailand/1B/2004	H5N1	PQ_RERRRKKR/G	6
A/pigeon/Thailand/Uttaradit-01/2004	H5N1	PQ_RERRRKKR/G	6
A/pigeon/Thailand/VSMU-7-NPT/2004	H5N1	PQ_RERRRKKR/G	6
A/pigeon/Samut Prakan/Thailand/CU-202/04	H5N1	PQ_RERRRKKR/G	6
A/pigeon/Thailand/VSMU-11-KRI/2005	H5N1	PQ_RERRRKKR/G	6
A/pigeon/Thailand/VSMU-13-KRI/2005	H5N1	PQ_RERRRKKR/G	6
A/pigeon/Thailand/VSMU-25-BKK/2005	H5N1	PQ_REKRRKKR/G	6
A/pigeon/Zhejiang/17/2005	H5N1	PQ_RERRRKR/G	5
A/pigeon/Jakarta/Hen406/2006	H5N1	PQ_RESRRKKR/G	5
A/pigeon/Nigeria/VRD370/2006	H5N1	PQ_GERRRKKR/G	6
A/Dove/Turkey/18/2006	H5N1	---	
A/Pigeon/Turkey/21/2006	H5N1	---	
A/pigeon/Turkey/Merkez 1334/2006	H5N1	PQ_GERRRKKR/G	6
A/pigeon/Turkey/Merkez 1090/2006	H5N1	PQ_GERRRKKR/G	6
A/pigeon/Turkey/Samsun ist2/2006	H5N1	PQ_GERRRKKR/G	6
A/pigeon/Laos/NCVD-36/2007	H5N1	PL_RERRRKR/G	5
A/pigeon/Laos/P0022/2007	H5N1	PL_RERRRKR/G	5
A/pigeon/Thailand/TS01/2007	H5N1	PQ_GERRRKKR/G	6
A/pigeon/Rostov-on-Don/6/2007	H5N1	PQ_GERRRKKR/G	6
A/Pigeon/Egypt/RIMD18/2009	H5N1	PQ_GERRRKKR/G	6
A/Pigeon/Egypt/RIMD20/2009	H5N1	PQ_GERRRKKR/G	6
A/feral pigeon/Hong Kong/3409/2009	H5N1	PQ_IERRRKKR/G	6
A/PIGEON/Hunan/185/2010	H5N1	PL_RERRRKR/G	5
A/pigeon/Huadong/QPG/2010	H5N1	PQ_IEGRRRKR/G	5
A/pigeon/Bhutan/01BR01/2012	H5N1	PQ_RERRRKR/G	5
A/pigeon/Hubei/RP25/2012	H5N1	PL_RERRRKR/G	5
A/pigeon/Egypt/Sharkia-2/2013	H5N1	---	
A/pigeon/Sichuan/NCXN29/2014	H5N1	PL_RERRRKR/G	5
A/pigeon/Zhejiang/112089/2014	H5N1	PQ_RERRRKR/G	5
A/pigeon/Zhejiang/112090/2014	H5N1	PQ_RERRRKR/G	5
A/pigeon/Zhejiang/727097/2014	H5N1	PQ_RERRRKR/G	5
A/pigeon/Egypt/Sharkia-1/2014	H5N1	PQ_GEKRRKKR/G	6
A/pigeon/Egypt/Sharkia-7/2014	H5N1	PQ_GEKRRKKR/G	6
A/pigeon/Egypt/Sharkia-9/2014	H5N1	---	
A/pigeon/Egypt/Sharkia-22/2014	H5N1	PQ_GEKRRKKR/G	6
A/pigeon/Egypt/HASH7/2015	H5N1	PK_GEKRRKKR/G	6
A/Pigeon/Egypt/ElSalom/2016	H5N1	PQ_GERRRKKR/G	6
A/pigeon/Egypt/HASHM3/2016	H5N1	PQ_GEKRRKKR/G	6
A/pigeon/Egypt/HASHM4/2016	H5N1	PK_GEKRRKKR/G	6
A/PIGEON/Egypt/A-Pigeon-1/2016	H5N1	PQ_GERRRKKR/G	6
A/pigeon/Egypt/SJCEIRR-RA19867OP/2021	H5N1	PL_REKRRKR/G	5
A/pigeon/Jiangxi/S40784/2021	H5N1	PL_REKRRKR/G	5
A/pigeon/Germany-NW/AI00951/2022	H5N1	PL_REKRRKR/G	5
A/Pigeon/Sweden/SVA220701SZ0238/FB002074/C-2022	H5N1	PL_REKRRKR/G	5
A/Columba_palumbus/Belgium/04695_0003/2023	H5N1	PL_REKRRKR/G	5
A/dove/Austria/23067602-001/2023	H5N1	PL_REKRRKR/G	5
A/pigeon/Texas/24-008765-001/2024	H5N1	PL_REKRRKR/G	5
A/wild pigeon/Jilin/CC01/2014	H5N6	PL_RERRRKR/G	5
A/Streptopelia decaocto/China/8/2016	H5N6	PQ_RERRRKR/G	5
A/pigeon/Fujian/1.17_FZHX0104-O/2017	H5N6	PL_RERRRKR/G	5
A/pigeon/Hainan/1.14_HKPL006-C/2018	H5N6	PL_RERRRKR/G	5
A/Speckled pigeon/South Africa/08-004B/2017	H5N8	PL_RERRRKR/G	5
A/pigeon/Cameroon/17RS1661-4/2017	H5N8	PL_REKRRKR/G	5
A/pigeon/South Africa/17080323/2017	H5N8	PL_REKRRKR/G	5
A/pigeon/Saudi Arabia/KFU-HKU3598_2017/2017	H5N8	PL_REKRRKR/G	5
A/pigeon/Saudi Arabia/KFU-HKU3599_2017/2017	H5N8	PL_REKRRKR/G	5
A/pigeon/Saudi Arabia/KFU-HKU3600_2017/2017	H5N8	PL_REKRRKR/G	5
A/pigeon/Egypt/A15052/2018	H5N8	PL_REKRRKR/G	5
A/pigeon/Egypt/A16800/2019	H5N8	PL_REKRRKR/G	5
A/pigeon/Egypt/A16804/2019	H5N8	PL_REKRRKR/G	5
A/pigeon/Egypt/A16805/2019(H5N8)	H5N8	PL_REKRRKR/G	5
A/pigeon/Kazakhstan/15-20-B-Talg-5/2020	H5N8	PL_REKRRKR/G	5
A/Columba_palumbus/Belgium/2928_004/2021	H5N8	PL_REKRRKR/G	5
A/pigeon/Egypt/SHAH-5803/2011	H5	PQ_GERRRKKR/G	6

## Data Availability

All previously published sequences of pigeon AIV strains were downloaded from the Global Initiative on Sharing Avian Influenza Data (GISAID, www.gisaid.org, accessed on 15 April 2025) and the Influenza Virus Resource at the National Center for Biotechnology Information (www.ncbi.nlm.nih.gov/genomes/FLU, accessed on 15 April 2025).

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
