# Peer review of "Detection of Avian Influenza Virus in Pigeons"

_viruses, 2025, doi:10.3390/v17040585_

Round 1
Reviewer 1 Report
Comments and Suggestions for Authors
This study gives a complete overview of multiple AIV subtypes circulating in pigeons, providing information about their distribution and genomic evolution. It will contribute to the understanding of the infection and prevalence situation of avian influenza virus in pigeons.
- line 59-62, “All previously published sequences of pigeon AIV strains were downloaded from the Global Initiative on Sharing Avian Influenza Data (GISAID, www.gisaid.org) and the Influenza Virus Resource at the National Center for Biotechnology Information(www.ncbi.nlm.nih.gov/genomes/FLU).” It should briefly state the cut-off date for the data acquisition from the database.
- line 59-62, What criteria were used to select the 238 HA sequences of the H6 subtype?
- The names of all strains included in the phylogenetic trees should be provided in the Supporting materials.
- From an evolutionary perspective, did these avian influenza viruses transiently spread from other poultry to the pigeon population, or have certain genotypes been continuously evolving within the pigeon population? Has the avian influenza virus in pigeons been transmitted to other animal species?
- The Figure 3 needs to be further optimized to make it more aesthetically pleasing and clear.
The English could be improved to more clearly express the research
Author Response
Comments 1: line 59-62, “All previously published sequences of pigeon AIV strains were downloaded from the Global Initiative on Sharing Avian Influenza Data (GISAID, www.gisaid.org) and the Influenza Virus Resource at the National Center for Biotechnology Information(www.ncbi.nlm.nih.gov/genomes/FLU).” It should briefly state the cut-off date for the data acquisition from the database.
Response 1: The cut-off date for the data acquisition from the database has been added in line 63 in the revised manuscript.
Comments 2: line 59-62, What criteria were used to select the 238 HA sequences of the H6 subtype?
Response 2: H6 genes of representative viruses were retrieved from public databases according to previous defied H6 groups (Huang 2010) as indicated in line 201.
Comments 3: The names of all strains included in the phylogenetic trees should be provided in the Supporting materials.
Response 3: The names of all strains included in the phylogenetic trees have been provided in the Supporting materials.
Comments 4: From an evolutionary perspective, did these avian influenza viruses transiently spread from other poultry to the pigeon population, or have certain genotypes been continuously evolving within the pigeon population? Has the avian influenza virus in pigeons been transmitted to other animal species?
Response 4: According to the suggestion, we have revised the title of the manuscript to “Detection of Avian Influenza Virus in pigeons”. We found that pigeons can be randomly infected with AIVs circulating in wild birds and poultry, therefore, the limited data for pigeon AIVs is insufficient to explain the evolution of the virus in pigeon population. In addition, we don't have enough evidence (including any references) to prove that AIV in pigeons been transmitted to other animal species.
Comments 5: The Figure 3 needs to be further optimized to make it more aesthetically pleasing and clear.
Response 5: Figure 3 has been optimized to make it more aesthetically pleasing and clear.
Reviewer 2 Report
Comments and Suggestions for Authors
In this manuscript, Cui and co-authors analyzed all available sequences of avian influenza viruses isolated from pigeons. These bird species are important players on the ecology of influenza A viruses, as they have close contacts with humans, wild birds and poultry. Therefore, there is no doubt that close surveillance of AIV in pigeons is required. This paper provides evolutionary analysis of the sequences isolated from pigeons, thus giving an impression on the susceptibility of these birds to different viral genotypes. The major point is that the title of the paper is somewhat misleading. Evolution of the virus within some hosts means that the virus is circulating among these birds and thus leading to some adaptive changes. Here, the authors state that pigeons can be randomly infected with AIVs circulating in wild birds and poultry, rather than transmitting viruses from pigeon to pigeon. So, it is better to change the title to one reflecting detection of various subtypes of avian influenza viruses in these species, rather than genetic evolution of the virus within this host.
Additional points:
- Please combine all Supplement tables into one table for better reading. It will be easier for readers to scroll through all viruses in one table rather than jumping from file to file.
- The quality of figures 1, 2, 4, 5 should be improved. Some inscriptions are difficult to read.
Please add the Discussion section before the Conclusion section, where some thoughts on the possible effects of transmission of AIVs from pigeons to poultry and humans, and vice versa (e.g. live bird markets) are discussed.
Author Response
Comments 1: The major point is that the title of the paper is somewhat misleading. Evolution of the virus within some hosts means that the virus is circulating among these birds and thus leading to some adaptive changes. Here, the authors state that pigeons can be randomly infected with AIVs circulating in wild birds and poultry, rather than transmitting viruses from pigeon to pigeon. So, it is better to change the title to one reflecting detection of various subtypes of avian influenza viruses in these species, rather than genetic evolution of the virus within this host.
Response 1: According to the suggestion, we have revised the title of the manuscript to “Detection of Avian Influenza Virus in pigeons”.
Comments 2: Please combine all Supplement tables into one table for better reading. It will be easier for readers to scroll through all viruses in one table rather than jumping from file to file.
Response 2: As recommended by the reviewer, all Supplement tables have been combined into one table.
Comments 3: The quality of figures 1, 2, 4, 5 should be improved. Some inscriptions are difficult to read.
Response 3: The quality of figures 1, 2, 4, 5 have been improved in the revised manuscript.
Comments 4: Please add the Discussion section before the Conclusion section, where some thoughts on the possible effects of transmission of AIVs from pigeons to poultry and humans, and vice versa (e.g. live bird markets) are discussed.
Response 4: We have combined the Results and discussion in our manuscript to better understand the results. In addition, we have discussed the possible effects of transmission of AIVs between pigeons and other hosts in every parts of the section.
Reviewer 3 Report
Comments and Suggestions for Authors
Cui et al. summarize and address the current relatively complete summary of the evolution and emergence of each subtypes of AIV in pigeons. In particular, they interrogate features of H5, H6, H7, H9 subtype pigeon AIV strains, which are relatively common in pigeons. Overall, this is an interesting study with reasonably convincing data to indicate that free-ranging pigeons always play a negligible, but non-zero role as the interspecies bridge in the ecology of influenza virus dynamics from the view point of the sequence analysis of pigeon AIV.
The pictures, in particular Figure 2 and Figure 5 are not clear and the text is blurred. In addition, the manuscript requires some reworking before publication can be considered. Specific points are listed below.
- Line 57: Please expand the Methods section with more detailed descriptions of the procedures, such as that data analysis should be detailed.
- Line 54, line 87, line 130, line 296 and elsewhere: ‘subtype’ should be used.
- Line 126-128: ‘The analysis of the sequence of A/pigeon/Guangxi/020P/2009(H3N6) indicated that the nucleotide sequences of both the HA and NA genes of this H3N6 strain belong to the Eurasian lineage’ sounds odd/nonsence.
- Line 138-139: ‘According to the amino acid sequence deduced from the HA gene segments’. The manuscript should clearly specify whether the sequence analysis was conducted de novo by the authors or derived from existing database resources. If the sequence analysis was performed by the authors, the Methods section must provide a comprehensive description of the analytical procedures, including software tools.
- Lines 300 to about 301. This sentence contains information that would be more suitable for the Introduction section. Or repositioned it to line 296. It describes nicely the substantial threats resulting from mixed AIV subtype infections.
Author Response
Comments 1: The pictures, in particular Figure 2 and Figure 5 are not clear and the text is blurred.
Response 1: The quality of figures have been improved in the revised manuscript.
Comments 2: Line 57: Please expand the Methods section with more detailed descriptions of the procedures, such as that data analysis should be detailed.
Response 2: Data analysis has been detailed in the Methods section.
Comments 3: Line 54, line 87, line 130, line 296 and elsewhere: ‘subtype’ should be used.
Response 3: As suggested by the reviewer, we have revised the language throughout the manuscript.
Comments 4: Line 126-128: ‘The analysis of the sequence of A/pigeon/Guangxi/020P/2009(H3N6) indicated that the nucleotide sequences of both the HA and NA genes of this H3N6 strain belong to the Eurasian lineage’ sounds odd/nonsence.
Response 4: As recommended by Reviewer, we have rephrased the sentence as follows:
“The analysis of the nucleotide sequence of A/pigeon/Guangxi/020P/2009(H3N6) indicated that both HA and NA genes belong to the Eurasian lineage.”
Comments 5: Line 138-139: ‘According to the amino acid sequence deduced from the HA gene segments’. The manuscript should clearly specify whether the sequence analysis was conducted de novo by the authors or derived from existing database resources. If the sequence analysis was performed by the authors, the Methods section must provide a comprehensive description of the analytical procedures, including software tools.
Response 5: The sequence analysis was conducted de novo by the authors, and data analysis procedures have been detailed in the Methods section.
Comments 6: Lines 300 to about 301. This sentence contains information that would be more suitable for the Introduction section. Or repositioned it to line 296. It describes nicely the substantial threats resulting from mixed AIV subtype infections.
Response 6: Following the reviewer's recommendation, we have repositioned this sentence to line 306 in the revised manuscript.